# Modes of Occurrence and Enrichment of Trace Elements in Coal from the Anjialing Mine, Pingshuo Mining District, Ningwu Coalfield, Shanxi Province, China

Tobechukwu Justin Ikeh [1], Beilei Sun [1,2], Chao Liu [1,2,*], Yunxia Liu [1], Yanlei Kong [1] and Xinyu Pan [1]

[1]  Department of Geosciences and Engineering, Taiyuan University of Technology, Taiyuan 030024, China
[2]  Shanxi Key Laboratory of Coal and Coal-Measure Gas Geology, Taiyuan 030024, China
*  Correspondence: liuchao06@tyut.edu.cn

**Abstract:** In this paper, the mineralogical composition, concentrations, distribution, and modes of occurrence of the trace elements in coal from the Anjialing coal seam 9 in the Pingshuo mining district, Ningwu coalfield, were studied using optical microscopy, X-ray powder diffraction (XRD), inductively coupled plasma optical emission spectrometry (ICP-OES), inductively coupled plasma mass spectrometry (ICP-MS), and sequential chemical extraction procedures (SCEPs). The identified minerals included mainly kaolinite, boehmite, pyrite, calcite, quartz, and muscovite. Compared to other hard coal from around the world, the coal from seam 9 was enriched with lithium (Li); slightly enriched with gallium (Ga), hafnium (Hf), zirconium (Zr), and mercury (Hg); typically enriched with lead (Pb), and depleted in arsenic (As). The results of the SCEPs analysis showed that Li, Ga, Zr, and Hf were mainly associated with clay minerals. Arsenic mainly occurred in its silicate and sulfide forms in pyrite and Pb was mainly associated with aluminosilicate, sulfide, and carbonate minerals.

**Keywords:** coal geochemistry; trace elements; enrichment; modes of occurrence; Anjialing coal mine

## 1. Introduction

Coal is the most abundant fossil fuel and provides a reliable and long-term fuel source for China and other countries, such as Turkey and South Africa [1,2]. As coal consumption in China increases, a considerable amount of pollution is generated, not only in gas emissions but also in ash residue.

The composition, enrichment, and origins of trace elements in coal have attracted a lot of attention worldwide [3–5], not only because of their essential geological information regarding basin evolution, regional tectonic history, and the deposition environment of coal-bearing strata [6–8], but also because of the potentially enormous economic benefits of the valuable elements (e.g., Li, Ga, U, Ge, Zr, Hf, and rare earth elements and yttrium (REY) [9,10] in coal or coal-bearing strata [11–13]. Dai et al. [14] studied the geochemical characteristics of trace elements in Chinese coal compared to those in other coal from around the world and found that the Chinese coal had typical background values for most trace elements, except for higher values of Li (31.8 µg/g), Zr (89.5 µg/g), Nb (9.44 µg/g), Ta (0.62 µg/g), Hf (3.71 µg/g), Th (5.84 µg/g), and rare earth elements ($\sum$La-Lu + Y = 136 µg/g). Shanxi plans to raise its annual coal output by 107 million metric tons to 1.3 billion tons in 2022 (https://english.www.gov.cn accessed on 2 July 2022), which accounted for more than a quarter of the total production of raw coal in China. Due to this massive coal production, the concentrations and origins of trace elements in coal from Shanxi Province have been extensively investigated over recent years. Previous studies have shown that some coalfields in Shanxi Province contain elevated levels of trace elements [15–18]. For instance, coal from the Laoyaogou mine in the Ningwu coalfield is rich in Li, Ga, Zr, Zn, Hf, Sr, Nb, Sn, Th, and rare earth elements and yttrium (REY); in particular,

the level of Li was found to be 163.42 μg/g, which is more than ten times greater than that found in other hard coal from around the world [19].

The modes of occurrence of an element in coal are important for dictating the behavior of the element during coal combustion, beneficiation, conversion, weathering, leaching, and any other chemical reaction that the coal undergoes [20]. This behavior of the element, in turn, determines how it impacts the environment, human health, technological performance, and byproduct recovery.

The modes of occurrence of the elements in coal can also provide insights into the sources of the element and some of the changes that occurred during coalification [21]. The importance of the modes of occurrence of trace elements in coal is attested by the number of published articles that have addressed this issue [22–24]. The elements in coal may be present in many different forms; for example, they can occur in silicates, sulfides, carbonates, oxides, phosphates, sulfates, phosphates, selenides, and halides. They may also occur on the surfaces of minerals or organic matter. Each of these forms could respond differently to individual solvents or the order of solvent application. Some minerals (particularly sulfides) are major hosts of toxic elements (such as As and Hg), which have adverse effects on human health (e.g., arsenonsis in Guizhou Province, southwestern China) and the environment [20]. In addition to the minerals themselves, harmful elements that are contained within the minerals can also contribute to human health issues [25].

This paper presents the results of a mineralogical and geochemical study of coal seam 9 in the Anjialing Mine, Pingshuo mining district, Ningwu coalfield, Shanxi Province. The mineralogical composition, concentrations, distribution, and modes of occurrence of trace elements in the coal from the Anjialing coal seam 9 were investigated using the following analysis methods: optical microscopy, X-ray powder diffraction (XRD), inductively coupled plasma optical emission spectrometry (ICP-OES), inductively coupled plasma mass spectrometry (ICP-MS), cold vapor atomic fluorescence spectrometry (CVAFS), and sequential chemical extraction procedures (SCEPs). The modes of occurrence of the trace elements in the area were examined using correlation analysis, cluster analysis, and sequential chemical extraction procedures. We hope that these research results will aid in the optimal utilization of coal.

## 2. Geological Setting

The Ningwu coalfield is located in the northern part of Shanxi Province and covers an area of about 2760 km$^2$, which is divided into four mining districts: Pingshuo, Shuonan, Xuangang, and Lanxian (Figure 1). The Ningwu basin is distributed in a NE–SW direction and is located in a special geographical position of geotectonic interest, with the Wutai Mountain anticline in the East, the Luya Mountain anticline in the west, the Sangganhe graben in the north, and the Luliang Mountain uplift in the southwest [26].

The Ningwu basin is dominated by a syncline structure that is composed of Paleozoic and Mesozoic strata. The bottom of the basin is made of an Early Precambrian metamorphic rock series and its core is composed of Jurassic strata. Triassic, Permian, Carboniferous, Ordovician, Cambrian, and Precambrian strata are exposed on both sides in turn (Devonian and Silurian strata are missing) and the occurrence of the strata is steep [27].

The Pingshuo mining district is located in the north of the Ningwu basin and is made up of late Cenozoic coal-bearing layers, including the Benxi formation, Taiyuan formation, and Shanxi formation. The Shanxi formation is composed of coal seams, pebbly gritstone, siltstone, and mudstone, which were deposited in a typical delta setting [28]. The Taiyuan formation is the primary coal-bearing stratum in the mining area and ranges from 35–115 m. The Anjialing mine is one of the surface coal mines in the area and includes coal seams 4, 9, and 11. Coal seam 9 is 2.4~36.5 m thick, with an average thickness of 13.45 m [29] (Figure 2).

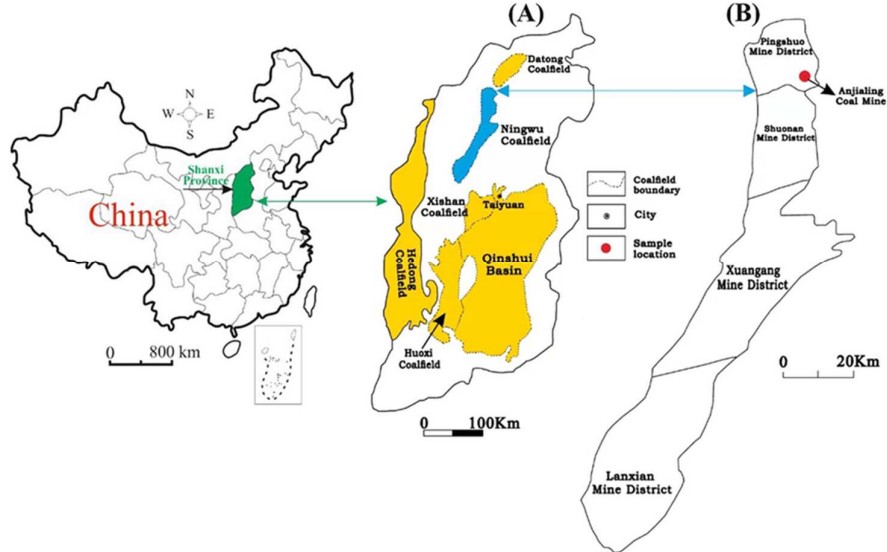

**Figure 1.** (**A**) The location of the Ningwu coalfield and the distribution of the major coalfields in Shanxi Province, China; (**B**) the Ningwu coalfield with the location of the Anjialing mine [30].

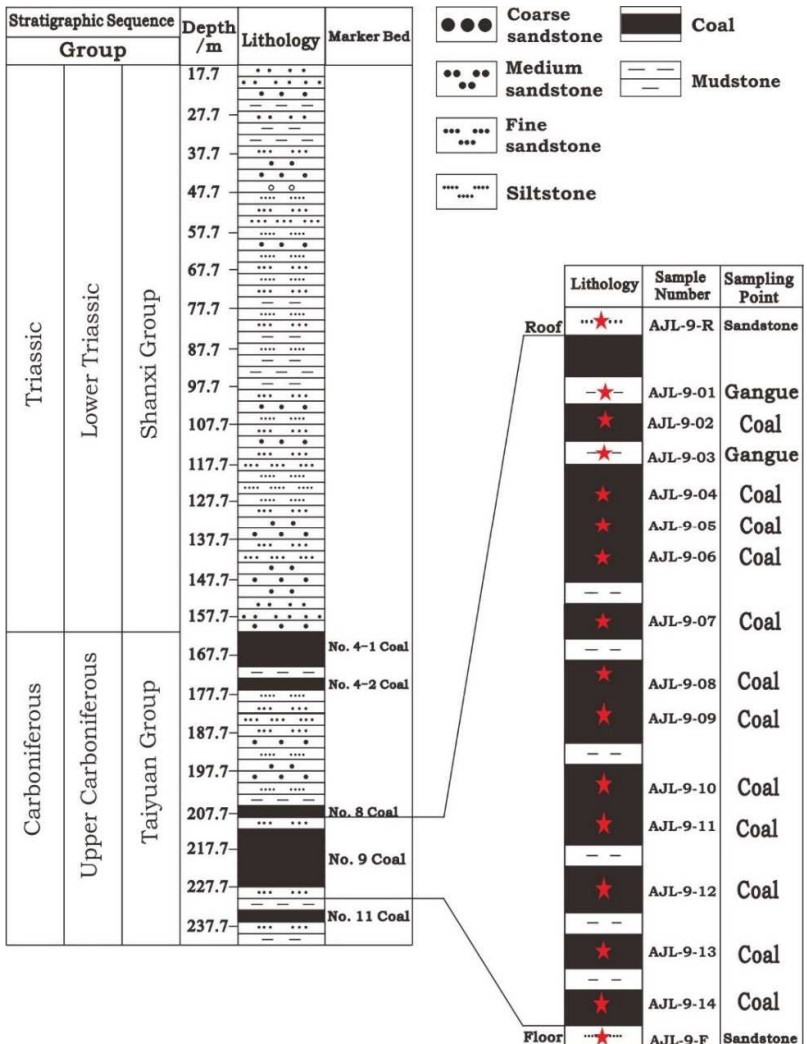

**Figure 2.** The stratigraphic sequence of the Anjialing coal mine and our sampling location.

### 3. Samples and Methods

The coal from the Anjialing coal seam 9 has an average volatile matter yield of 41.15%, with an average ash yield of 22.95%. The coal has an average moisture content of 2.46%. This coal also has an average total sulfur level of 2.17%, which is mostly organic (an average of 1.71%), with an average sulfide sulfur content of 0.65% and an average sulfate sulfur content of 0.04% [31]. The coal has an average calorific capacity value of 24.60 MJ/kg [31]. The vitrinite random reflectance of the coal varies from 0.532% to 0.659% (0.60% on average), which indicates that this coal is a highly volatile bituminous coal [32].

A total of sixteen samples (twelve coal, two gangues, one roof, and one floor) were collected from a borehole in the study area. From top to bottom, the samples were labeled as Roof, AJL-9-01G, AJL-9-02, AJL-9-03G, AJL-9-04 to AJL-9-14, and Floor (the gangue samples were identified with the suffix "G" and the roof and floor samples were sandstone). All of the collected samples were immediately wrapped and stored in plastic bags to minimize contamination and oxidation.

The original samples were freeze-dried and divided into two halves: one half was used for analyses, while the other half was kept as a reserve in a storage facility. For sample processing, we used a sample crushing machine, mortar, sample sieves, and an electronic balance.

An XRD analysis (Ultima IV ray diffractometer with Cu and a scintillation detector, which were developed by Rigaku, Tokyo, Japan) that recorded XRD over a 2θ interval of 5–80° with a step increment of 0.02° and optical microscopy (Leica DM 2500P microscope by Leica Microsystems, Solms, Germany) were used to determine the mineral contents and compositions of the twelve coal samples (AJL-9-02 and AJL-9-04 to AJL-9-14). ICP-OES was carried out on the twelve coal samples to determine their major elements (Agilent 720, Agilent, Santa Clara, CA, USA). The contents of trace elements in the coal were determined using ICP-MS (Agilent7900). $SiO_2$ was digested in the samples using the alkali melting method, in which 0.05 g of a sample was dissolved in a crucible with 0.25 g of NaOH, heated at 700 °C for 30 min, dissolved with hot water after cooling, and acidified with 5 mL of HCL before being added to 250 mL for ICP-OES determination. The mercury content in the coal samples was determined using CVAFS (particle size < 200).

In this study, the petrography of the coal was also determined. The coal samples AJL-9-2 and AJL-9-4 to AJL-9-14 were made into coal bricks by mixing the ground coal sample with a binder in a ratio of 2:1. After mixing evenly, the mixture was put into the crucible, the heat was applied, and it was constantly stirred. The mixture was then quickly placed into an inlay machine at a pressure of about 3.5 mpa for about 30 s. We then removed the coal bricks and numbered them. Next, the coal bricks were further ground down. When grinding, the bricks needed to be held firm by hand and rotated in the opposite direction of the rotary table with a little applied pressure. The grinding continued until the surfaces of the coal bricks were smooth, without any scratches or luster, and the boundaries of the coal particles were clear. The surfaces of the bricks were polished evenly to meet the following requirements: flat and smooth surfaces with no protrusions, dents, obvious scratches, stains or abrasions and the microscopic compositions of the coal could be observed.

In this study, the researchers carried out step-by-step chemical extraction procedures on four coal samples (AJL-9-07, AJL-9-08, AJL-9-10, and AJL-9-13) to evaluate the modes of occurrence of the trace elements in the coal, as presented in Table 1.

**Table 1.** The results of the six-step sequential chemical extraction procedure (SCEP).

| Step | Modes of Occurrence | Method in Each Step |
|------|---------------------|---------------------|
| 1 | Water Soluble State | 1 g of coal sample + 1 mL of ethanol + 20 mL of water at normal temp. for 2 h, centrifuge for 30 min at 4000/r, and then add 20 mL of water and centrifuge for 30 min at 4000/r<br>↓ |

| Step | Modes of Occurrence | Method in Each Step |
|------|--------------------|--------------------|
| 2 | Ion-Exchangeable Bound State | Residue + 20 mL 1 mol/L of $CH_3COONH_4$ at 40 °C for 2 h, centrifuge for 30 min at 4000/r, and then add 20 mL of water and centrifuge for 30 min at 4000/r (three times) |
| | | ↓ |
| 3 | Carbonate Bound State | Residue + 20 mL 1 mol/L of $CH_3COOH$ at 40 °C for 2 h, centrifuge for 30 min at 4000/r, and then add 20 mL of water and centrifuge for 30 min at 4000/r (three times) |
| | | ↓ |
| 4 | Sulfide Bound State | Residue + 20 mL 4 mol/L of $HNO_3$ at 80 °C for 2 h, centrifuge for 30 min at 4000/r, and then add 20 mL of water and centrifuge for 30 min at 4000/r (three times) |
| | | ↓ |
| 5 | Organic Bound State | Residue + 15 mL 0.02 mol/L of $HNO_3$ + 5 mL of 30% $H_2O_2$ at 60 °C for 3 h, add 5 mL of 30% $H_2O_2$ at 60 °C for 3 h, add 5 mL 3.2 mol/L of $CH_3COONH_4$ at normal temp. for 3 h, centrifuge, and then add 20 mL of water and centrifuge for 30 min at 4000/r (three times) |
| | | ↓ |
| 6 | Silicate Bound State | Residue |

## 4. Results and Discussion

### 4.1. Mineralogical Analysis

The compositions of minerals in the coal samples were determined using XRD analysis and optical microscopy. The results showed that the minerals in the coal from the Anjialing coal seam 9 mainly consisted of kaolinite, pyrite, and quartz, with occasional calcite, illite, and boehmite (Figure 3).

Under the reflected light of oil immersion, the clay minerals in the coal samples appeared brownish. Kaolinite was observed under reflected light and mainly occurred in the infillings of cell cavities in fusinite (Figure 4A,C). The carbonate minerals in the coal were mainly dominated by calcite. Calcite was commonly observed under the microscope in the thin sections of coal from the Anjialing coal seam 9. It could be seen that the calcite in the coal formed directionally along fractured surfaces as infillings in the coal (Figure 4B).

The sulfide minerals in the coal were mainly pyrite. The formation of pyrite is affected by space and environment and it has different forms, some of which fill cell cavities and fissures. The pyrite crystals in our samples were brass-colored (bright yellow) under reflected light (Figure 4D).

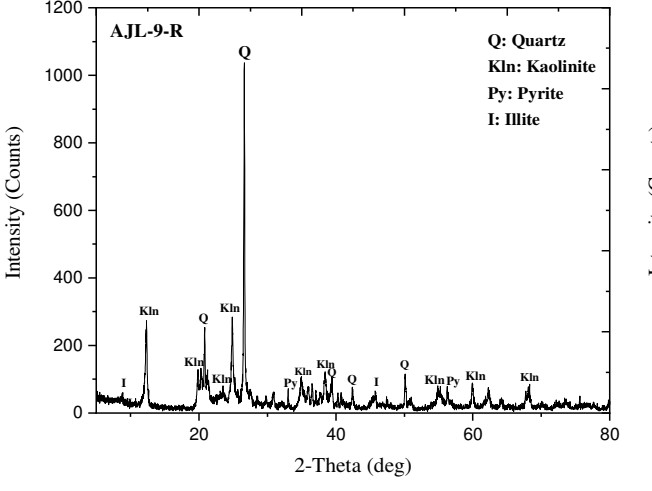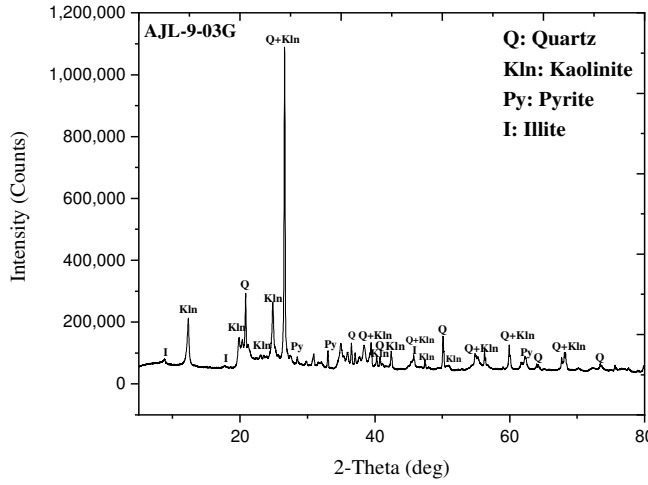

**Figure 3.** *Cont.*

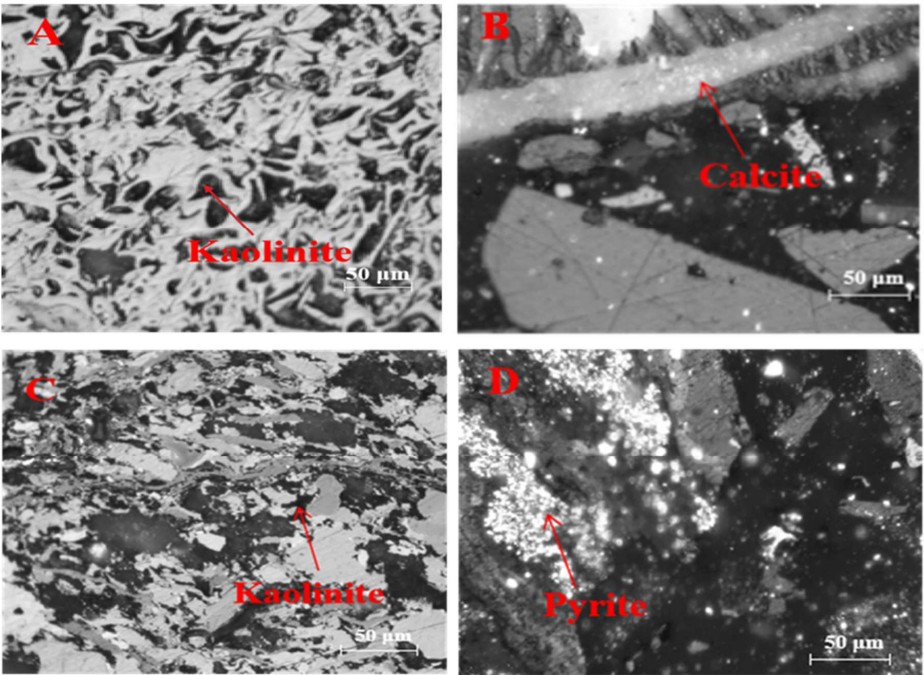

**Figure 3.** The XRD patterns of the mineral components in coal from the Anjialing coal seam 9.

**Figure 4.** The minerals in the coal samples: (**A**) kaolinite; (**B**) calcite; (**C**) kaolinite; (**D**) pyrite (under reflected light).

### 4.2. Concentrations and Distribution of Elements

4.2.1. Major Element Oxides

The proportions of the major elements in the coal from the Anjialing coal seam 9 are listed in Table 2. Compared to the average values in Chinese coal, the Anjialing coal contained a more significant average percentage of $Al_2O_3$ and lower average percentages of $SiO_2$ and $TiO_2$ due to the high contents of kaolinite and boehmite (Figure 3). The proportions of the remaining major elements were lower than the respective average values in Chinese coal. The average $SiO_2/Al_2O_3$ ratio of our coal samples (2.42) and gangue (2.78) were significantly higher than the average values in Chinese coal (1.42) [14], which could be due to the presence of quartz in the coal samples. The major elements in the coal from the Anjialing coal seam 9 were dominated by $SiO_2$ and $Al_2O_3$, which indicated that quartz and clay minerals were the principal carriers of these elements.

**Table 2.** The contents of major element oxides in the coal from the Anjialing coal seam 9 (%).

| Sample | Type | $Al_2O_3$ | $SiO_2$ | $Na_2O$ | $MgO$ | $P_2O_5$ | $K_2O$ | $CaO$ | $TiO_2$ | $MnO$ | $Fe_2O_3$ | $SiO_2/Al_2O_3$ |
|---|---|---|---|---|---|---|---|---|---|---|---|---|
| AJL-9-R | Sandstone | 26.99 | 51.43 | 0.05 | 0.47 | 0.11 | 1.05 | 0.68 | 1.02 | 0.09 | 4.310 | 1.91 |
| AJL-9-01G | Gangue | 20.27 | 59.10 | 0.10 | 1.24 | 0.10 | 2.68 | 1.14 | 0.72 | 0.03 | 2.511 | 2.92 |
| AJL-9-02 | Coal | 2.44 | 6.11 | 0.004 | 0.06 | 0.008 | 0.012 | 1.5 | 0.17 | 0.008 | 0.279 | 2.50 |
| AJL-9-03G | Gangue | 21.21 | 56.18 | 0.07 | 1.06 | 0.14 | 2.13 | 0.88 | 0.73 | 0.03 | 3.208 | 2.65 |
| AJL-9-04 | Coal | 5.40 | 10.32 | 0.012 | 0.37 | 0.018 | 0.023 | 4.09 | 0.15 | 0.014 | 0.453 | 1.91 |
| AJL-9-05 | Coal | 5.42 | 9.56 | 0.008 | 0.12 | 0.012 | 0.02 | 1.04 | 0.28 | 0.002 | 0.315 | 1.76 |
| AJL-9-06 | Coal | 6.89 | 22.02 | 0.02 | 0.07 | 0.018 | 0.16 | 0.13 | 0.54 | 0.000 | 0.156 | 3.2 |
| AJL-9-07 | Coal | 6.96 | 22.32 | 0.024 | 0.10 | 0.03 | 0.2 | 0.38 | 0.34 | 0.001 | 0.188 | 3.21 |
| AJL-9-08 | Coal | 11.61 | 16.76 | 0.014 | 0.08 | 0.02 | 0.04 | 2.94 | 0.26 | 0.010 | 0.184 | 1.44 |
| AJL-9-09 | Coal | 5.42 | 9.73 | 0.006 | 0.06 | 0.03 | 0.024 | 2.05 | 0.22 | 0.005 | 0.102 | 1.8 |
| AJL-9-10 | Coal | 5.23 | 25.46 | 0.013 | 0.03 | 0.019 | 0.064 | 0.48 | 0.40 | 0.001 | 0.133 | 4.87 |
| AJL-9-11 | Coal | 4.18 | 8.13 | 0.005 | 0.04 | 0.013 | 0.025 | 0.49 | 0.18 | 0.001 | 0.745 | 1.95 |
| AJL-9-12 | Coal | 3.18 | 6.72 | 0.003 | 0.04 | 0.16 | 0.02 | 0.26 | 0.14 | 0.001 | 0.457 | 2.11 |
| AJL-9-13 | Coal | 4.11 | 11.55 | 0.004 | 0.04 | 0.02 | 0.05 | 0.18 | 0.32 | 0.001 | 1.531 | 2.81 |
| AJL-9-14 | Coal | 3.4 | 6.78 | 0.002 | 0.05 | 0.008 | 0.014 | 0.623 | 0.14 | 0.002 | 0.407 | 2.00 |
| AJL-9-F | Sandstone | 29.32 | 51.43 | 0.05 | 0.26 | 0.03 | 0.19 | 0.17 | 0.74 | 0.010 | 1.470 | 1.75 |
| AV-C | | 5.35 | 12.96 | 0.01 | 0.09 | 0.03 | 0.05 | 1.18 | 0.26 | 0.000 | 0.410 | 2.42 |
| AV-G | | 20.74 | 57.64 | 0.08 | 1.15 | 0.123 | 2.40 | 1.009 | 0.73 | 0.03 | 2.859 | 2.78 |
| AV-SST | | 28.16 | 51.38 | 0.05 | 0.37 | 0.07 | 1.08 | 0.39 | 0.88 | 0.05 | 2.890 | 1.83 |
| China [a] | | 5.98 | 8.47 | 0.16 | 0.22 | 0.09 | 0.19 | 1.23 | 0.33 | 0.02 | 4.850 | 1.42 |

[a] Average contents of major element oxides in Chinese coal, from Dai et al. (2012) [14]. AV-C, average concentration of coal samples; AV-G, average concentration of gangue samples; AV-SST, average concentration of sandstone samples.

4.2.2. Trace Elements

The concentration coefficient (CC) was derived as the ratio of the concentration of a certain trace element in the analyzed samples to the average value of that trace element in other hard coal from around the world. The average contents of trace elements in the Anjialing coal compared to those of other hard coal from around the world are presented in Table 3 [33]. According to the CC classification levels (enrichment (CC > 5), slight enrichment (2 < CC < 5), normal (0.5 < CC < 2) or low (CC < 0.5)) that were proposed by Dai et al. [34], the average concentration of Li in the Anjialing coal was more than five times greater than that in other hard coal from around the world with a concentration coefficient of 5.81 (Figure 5). Gallium, Zr, Hf, U, and Hg were also abundant in our coal samples and their enrichment coefficients were 2.53, 3.64, 2.78, 2.06, and 3.80, respectively. However, cobalt, Ni, As, Rb, Cs, Ba, Ge, Sb, TI, and Bi levels were depleted and their concentration coefficients were 0.31, 0.24, 0.1, 0.11, 0.14, 0.20, 0.23, 0.3, 0.22, and 0.33 respectively (CC < 0.5). The remaining elements (Be, Sc, V, Cr, Zn, Sr, Y, Nb, Mo, La, Ce, Pr, Nd, Sm, Eu, Gd, Tb, Dy, Ho, Er, Tm, Yb, Lu, Ta, Pb, Th, and W) had similar concentrations (0.5 < CC < 2) in the Anjialing samples to the average contents in other hard coal from around the world (Table 3 and Figure 5).

**Table 3.** The concentrations of trace elements (µg/g) in the coal from the Anjialing coal seam 9 and those in other hard coal from around the world.

| Sample | AJL-9-R | AJL-9-01G | AJL-9-02 | AJL-9-03G | AJL-9-04 | AJL-9-05 | AJL-9-06 | AJL-9-07 | AJL-9-08 | AJL-9-09 | AJL-9-10 | AJL-9-11 | AJL-9-12 | AJL-9-13 | AJL-9-14 | AJL-9-F | [a] World Coal | [b] Coal | AV-C | AV-G | CC |
|---|---|---|---|---|---|---|---|---|---|---|---|---|---|---|---|---|---|---|---|---|---|
| Li | 11.7 | 42.3 | 19.3 | 46.2 | 49.0 | 59.9 | 113 | 150 | 121 | 87.1 | 181 | 66.4 | 40.8 | 60.9 | 27.0 | 74.0 | 14.0 | 14.0 | 81.3 | 44.3 | 5.81 |
| Be | 1.41 | 2.84 | 3.61 | 3.01 | 1.48 | 1.51 | 1.05 | 1.11 | 1.17 | 1.21 | 1.17 | 1.47 | 3.36 | 1.94 | 1.95 | 1.68 | 2.00 | 2.00 | 1.75 | 2.93 | 0.88 |
| Sc | 8.64 | 7.62 | 2.62 | 7.32 | 6.03 | 1.81 | 0.59 | 0.74 | 3.23 | 3.37 | 0.23 | 0.83 | 1.23 | 1.04 | 5.87 | 5.65 | 3.70 | 49.0 | 2.30 | 7.47 | 0.62 |
| V | 94.9 | 115 | 37.7 | 156 | 25.5 | 14.0 | 29.6 | 23.7 | 8.60 | 15.7 | 15.1 | 7.09 | 13.8 | 26.0 | 18.7 | 28.2 | 28.0 | 21.0 | 19.6 | 135.5 | 0.70 |
| Cr | 47.0 | 62.0 | 27.6 | 78.7 | 21.0 | 10.0 | 14.9 | 11.6 | 11.0 | 13.5 | 9.58 | 7.93 | 14.1 | 24.6 | 21.4 | 13.5 | 17.0 | 12.0 | 15.6 | 70.4 | 0.92 |
| Co | 13.1 | 15.3 | 2.79 | 18.1 | 4.60 | 1.53 | 1.58 | 1.97 | 1.22 | 1.35 | 1.23 | 0.74 | 1.33 | 2.33 | 1.51 | 4.74 | 6.00 | 7.00 | 1.85 | 16.7 | 0.31 |
| Ni | 33.4 | 40.2 | 6.86 | 45.4 | 17.1 | 1.63 | 1.76 | 0.98 | 3.85 | 2.67 | 0.00 | 0.00 | 0.00 | 1.32 | 0.48 | 20.4 | 17.0 | 14.0 | 4.07 | 42.8 | 0.24 |
| Cu | 66.6 | 31.3 | 18.0 | 36.9 | 20.8 | 12.0 | 26.9 | 26.3 | 13.2 | 13.8 | 14.0 | 11.4 | 16.4 | 15.7 | 11.4 | 99.8 | 16.0 | 13.0 | 16.7 | 34.1 | 1.04 |
| Zn | 132 | 113 | 22.0 | 143 | 19.1 | 40.0 | 25.7 | 18.4 | 13.8 | 13.9 | 16.9 | 12.5 | 18.9 | 22.6 | 24.4 | 113 | 28.0 | 35.0 | 20.7 | 126 | 0.74 |
| Ga | 20.6 | 23.0 | 6.43 | 22.6 | 35.0 | 18.7 | 11.9 | 14.4 | 12.4 | 20.5 | 9.89 | 7.89 | 13.5 | 13.7 | 18.1 | 17.9 | 6.00 | 9.00 | 15.2 | 22.8 | 2.53 |
| As | 4.07 | 11.4 | 0.85 | 16.9 | 0.37 | 0.37 | 0.07 | 0.36 | 0.46 | 0.50 | 0.70 | 1.10 | 0.77 | 4.33 | 1.29 | 1.83 | 9.00 | 5.00 | 0.93 | 14.2 | 0.10 |
| Rb | 39.9 | 138 | 0.85 | 110 | 1.15 | 0.69 | 5.44 | 6.20 | 1.72 | 1.01 | 2.74 | 1.22 | 0.72 | 2.51 | 0.66 | 31.1 | 18.0 | 8.00 | 2.05 | 124 | 0.11 |
| Sr | 82.2 | 124 | 115 | 134 | 186 | 119 | 76.1 | 131 | 187 | 242 | 88.9 | 89.0 | 821 | 101 | 143 | 39.4 | 100 | 136 | 191.6 | 129 | 1.92 |
| Zr | 388 | 190 | 52.6 | 175 | 131 | 127 | 223 | 188 | 84.8 | 114 | 177 | 66.6 | 143 | 136 | 129 | 402 | 36.0 | 52.0 | 131 | 182.5 | 3.64 |
| Nb | 18.8 | 15.1 | 5.19 | 15.6 | 9.04 | 9.63 | 0.84 | 9.48 | 9.73 | 11.1 | 12.5 | 3.72 | 5.46 | 9.63 | 7.32 | 16.3 | 4.00 | 14.0 | 7.80 | 30.7 | 1.95 |
| Mo | 1.72 | 0.95 | 2.31 | 1.14 | 2.53 | 1.58 | 0.97 | 1.64 | 1.42 | 1.79 | 1.35 | 1.01 | 1.48 | 1.68 | 1.21 | 1.11 | 2.10 | 4.00 | 1.58 | 1.04 | 0.75 |
| Cs | 2.51 | 5.50 | 0.08 | 5.96 | 0.08 | 0.08 | 0.19 | 0.21 | 0.24 | 0.09 | 0.23 | 0.15 | 0.10 | 0.24 | 0.065 | 1.55 | 1.10 | 1.00 | 0.15 | 5.73 | 0.14 |
| Ba | 148 | 346 | 32.7 | 300 | 57.1 | 19.1 | 28.8 | 41.9 | 41.9 | 47.9 | 21.5 | 15.6 | 32.6 | 17.6 | 10.4 | 141 | 150 | 82.0 | 30.6 | 323 | 0.20 |
| Hf | 10.6 | 5.78 | 1.28 | 5.49 | 3.07 | 3.41 | 5.93 | 4.19 | 2.57 | 2.92 | 4.27 | 1.93 | 3.76 | 3.67 | 3.10 | 10.8 | 1.20 | 2.40 | 3.34 | 5.64 | 2.78 |
| Ta | 1.09 | 1.02 | 0.23 | 1.03 | 0.30 | 0.67 | 0.04 | 0.46 | 0.87 | 0.69 | 0.92 | 0.07 | 0.28 | 0.53 | 0.33 | 1.03 | 0.30 | 0.70 | 0.45 | 1.03 | 1.50 |
| Pb | 26.7 | 32.9 | 6.58 | 21.4 | 16.0 | 23.6 | 43.6 | 25.9 | 10.7 | 14.2 | 21.4 | 15.0 | 10.4 | 17.1 | 8.33 | 21.9 | 9.00 | 13.0 | 17.73 | 27.2 | 1.97 |
| Th | 13.6 | 17.2 | 5.48 | 17.7 | 6.16 | 7.37 | 3.35 | 3.88 | 10.8 | 7.53 | 1.86 | 6.01 | 5.37 | 6.94 | 6.50 | 5.41 | 3.20 | 6.00 | 5.94 | 17.5 | 1.86 |
| U | 4.00 | 3.30 | 5.96 | 3.62 | 7.97 | 4.39 | 2.96 | 3.83 | 3.12 | 3.70 | 2.09 | 1.68 | 2.76 | 3.96 | 4.51 | 1.47 | 1.90 | 3.00 | 3.91 | 3.46 | 2.06 |
| Ge | | 2.02 | 1.01 | 2.40 | 0.99 | 0.42 | 0.14 | 0.23 | 0.24 | 0.17 | 0.13 | 0.67 | 0.69 | 1.27 | 0.78 | | 2.40 | 2.78 | 0.56 | 2.21 | 0.23 |
| Cd | | 14.3 | 53.9 | 175.6 | 40.2 | 40.3 | 43.2 | 7.76 | 91.8 | 28.2 | 17.6 | 66.9 | 77.2 | 24.4 | 18.3 | | 0.20 | 0.20 | 42.5 | 94.9 | 212.4 |
| Sb | | 0.43 | 0.17 | 1.31 | 0.53 | 0.38 | 0.24 | 0.32 | 0.32 | 0.30 | 0.32 | 0.25 | 0.27 | 0.35 | 0.18 | | 1.00 | 2.00 | 0.30 | 0.87 | 0.30 |
| W | | 1.26 | 0.73 | 1.46 | 0.94 | 1.09 | 0.06 | 0.81 | 1.82 | 1.15 | 0.92 | 0.14 | 0.79 | 0.72 | 0.98 | | 0.99 | 2.00 | 0.85 | 1.36 | 0.86 |
| Tl | | 0.53 | 0.31 | 0.37 | 0.04 | 0.04 | 0.07 | 0.05 | 0.06 | 0.03 | 0.04 | 0.23 | 0.06 | 0.54 | 0.11 | | 0.58 | 0.40 | 0.13 | 0.45 | 0.22 |
| Bi | | 0.39 | 0.29 | 0.38 | 0.31 | 0.23 | 0.51 | 0.60 | 0.48 | 0.43 | 0.45 | 0.27 | 0.27 | 0.27 | 0.18 | | 1.10 | 0.80 | 0.36 | 0.39 | 0.33 |
| Hg | | 0.31 | 0.36 | 0.29 | 0.15 | 0.31 | 0.21 | 0.23 | 0.27 | 0.28 | 0.73 | 0.54 | 0.58 | 0.62 | 0.33 | | 0.10 | 0.15 | 0.38 | 0.30 | 3.80 |

[a] World Coal averages values in hard coal from around the world from Ketris and Yudovich (2009) [33]; [b] Coal, average values in Chinese coal from Dai et al. (2012) [14]; AV-C, average concentration of coal samples; AV-G, average concentration of gangue samples; CC, concentration coefficient = AV-C/World Coal.

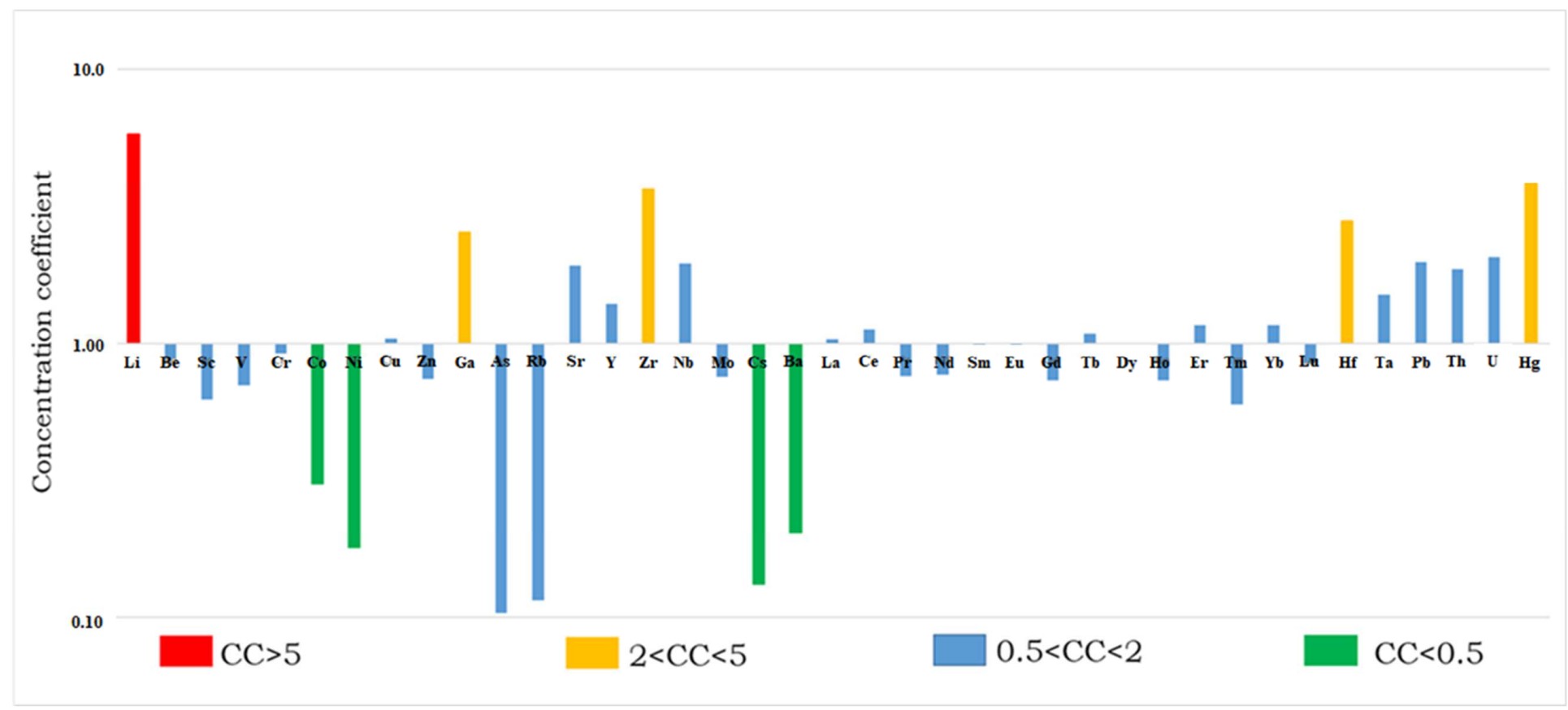

**Figure 5.** The concentration coefficients (CCs) of the trace elements and REY in the Anjialing coal samples.

*4.3. Distribution Characteristics of Selected Trace Elements*

Figure 6 shows the vertical variation trends of the trace elements in the coal samples. The Li contents in the samples were higher toward the middle of the seam (AJL-9-06 to AJL-9-10), while the Li contents appeared to be lower toward the roof and floor. The Ga contents were higher in AJL-9-01G, AJL-9-03G, AJL-9-04, and AJL-9-09 than in the other samples. The vertical variations in Hf and Zr were very similar, which suggested that these elements could be from the same source region or origin. The highest contents of arsenic were in the gangue samples, which indicated that the gangue contained a more significant amount of As. The Pb contents were higher in AJL-9-01G and toward the middle of the seam (AJL-9-03G to AJL-9-07) and appeared to be lower toward the floor. The Hg contents were higher toward the floor (AJL-9-10 to AJL-9-13) and appeared to be lower toward the roof.

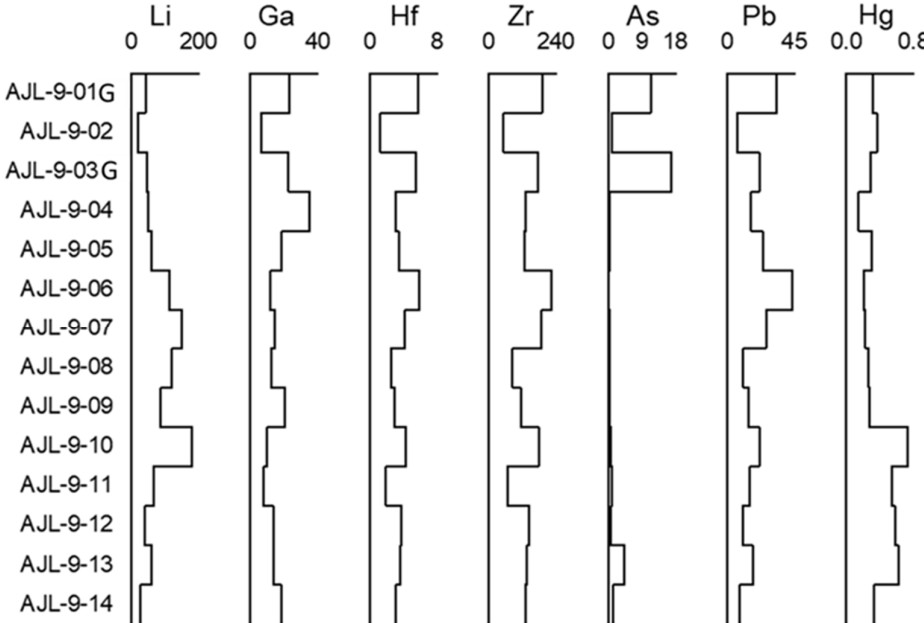

**Figure 6.** The vertical distribution of selected trace elements in the coal from the Anjialing coal seam 9.

## 5. Modes of Occurrence and Enrichment of Selected Trace Elements in Coal

*5.1. Sequential Chemical Extraction Procedures*

Sequential chemical extraction procedures (SCEPs, Table 1) were also employed to study the modes of occurrence of Li, Ga, As, Zr, and Pb in four coal samples. However, the modes of occurrence of these trace elements were divided into the following states: water soluble, ion-exchangeable, carbonate, sulfide, organic, and silicate. Table 4 shows the proportions of the selected trace elements (Li, Ga, As, Zr, and Pb) in the Anjialing coal samples.

In the Anjialing coal samples, Li occurred mainly in silicates. The proportion of lithium in silicates accounted for more than 70% of the Li in AJL-9-7, AJL-9-10, and AJL-9-13, followed by water soluble (8%–9%) and organic (2%–5%) Li. Li occurred in the sulfide state more in AJL-9-7 than in the other samples. Lithium occurred less often in the ion-exchangeable and carbonate states in all of the samples.

In AJL-9-8 and AJL-9-10, Ga occurred mainly in the residue (such as silicate). The proportion of residue in the samples was >50%, followed by the sulfide state in AJL-9-7 (41%) and AJL-9-13 (39%), which was lower in AJL-9-8 and AJL-9-10. Ga occurred less often in the water soluble (3%) and organic (1%–4%) states.

**Table 4.** Modes of occurrence of Li, Ga, As, Zr, and Pb in the Anjialing coal samples.

| | Li Proportion (%) | | | | | |
|---|---|---|---|---|---|---|
| Sample | Silicate Form | Organic Form | Sulfide Form | Carbonate Form | Ion-Exchangeable Form | Water Soluble Form |
| AJL-9-7 | 74 | 2 | 11 | 4 | 0 | 9 |
| AJL-9-8 | 82 | 4 | 0 | 5 | 1 | 8 |
| AJL-9-10 | 82 | 5 | 1 | 4 | 1 | 9 |
| AJL-9-13 | 82 | 2 | 3 | 1 | 0 | 12 |
| | Ga Proportion (%) | | | | | |
| AJL-9-7 | 55 | 1 | 41 | 0 | 0 | 3 |
| AJL-9-8 | 94 | 2 | 0 | 1 | 0 | 3 |
| AJL-9-10 | 94 | 2 | 0 | 1 | 0 | 3 |
| AJL-9-13 | 57 | 4 | 39 | 0 | 0 | 0 |
| | Zr Proportion (%) | | | | | |
| AJL-9-7 | 47 | 17 | 37 | 0 | 0 | 0 |
| AJL-9-8 | 70 | 29 | 1 | 0 | 0 | 0 |
| AJL-9-10 | 66 | 34 | 0 | 0 | 0 | 0 |
| AJL-9-13 | 56 | 22 | 22 | 0 | 0 | 0 |
| | As Proportion (%) | | | | | |
| AJL-9-7 | 92 | 0 | 7 | 0 | 0 | 1 |
| AJL-9-8 | 98 | 0 | 0 | 0 | 0 | 2 |
| AJL-9-10 | 98 | 0 | 0 | 0 | 0 | 2 |
| AJL-9-13 | 35 | 2 | 62 | 0 | 0 | 2 |
| | Pb Proportion (%) | | | | | |
| AJL-9-7 | 3 | 1 | 86 | 8 | 1 | 0 |
| AJL-9-8 | 13 | 7 | 15 | 49 | 6 | 9 |
| AJL-9-10 | 11 | 5 | 8 | 63 | 10 | 2 |
| AJL-9-13 | 4 | 1 | 85 | 8 | 4 | 2 |

Zirconium occurred mainly in the residue (such as silicate). The proportion of residue in the samples was >40%, followed by the sulfide state in AJL-9-7 (37%) and AJL-9-13 (22%) and the organic state (17%–34%).

In the coal samples, silicates were the most significant carriers of As because the proportion of Li in the silicate fractions was more than 90% in AJL-9-7, AJL-9-8, and AJL-9-10, followed by the sulfide fractions in AJL-9-7 (7%) and AJL-9-13 (62%). Notably, As occurred in both the silicate and sulfide fractions in the AJL-9-13 sample.

The mode of occurrence of Pb was mainly related to the sulfide state and had an elemental proportion of more than 80% in AJL-9-07 and AJL-9-13, whereas carbonates uncommonly played an important role in Pb occurrence in the other two samples. Lead also occurred in the silicate, ion-exchangeable (1%–10%), organic (1%–7%), and water soluble (2%–9%) states in a decreasing sequence.

### 5.2. Geochemical Associations in the Coal Samples

The modes of occurrence of the trace elements in the Anjialing coal samples were investigated using cluster analysis and correlation coefficients, which are practical indirect approaches for examining coal geochemistry [35]. The hierarchical clustering was based on Pearson correlation coefficients. The most closely correlated elements and major elements are linked first, followed by the elements or groups of elements with decreasing correlations until a complete dendrogram is obtained, as described in Zhao et al. [36].

According to the R cluster analysis results, which were based on the correlation matrix, the elements were grouped step-by-step in a hierarchical analysis procedure (Figure 7) of 48 trace elements and 10 major elements. The modes of occurrence of the trace elements

could be inferred from the results of the R cluster analysis and the correlation coefficients. The clustering method was used to identify the connections between groups and the interval correlations were calculated according to the Pearson correlation coefficients and were normalized to a maximum value of 1, which resulted in eight clusters at a point distance level of 15 (Figure 7) [37]. However, the authors considered four groups with elements of interest, which are detailed below.

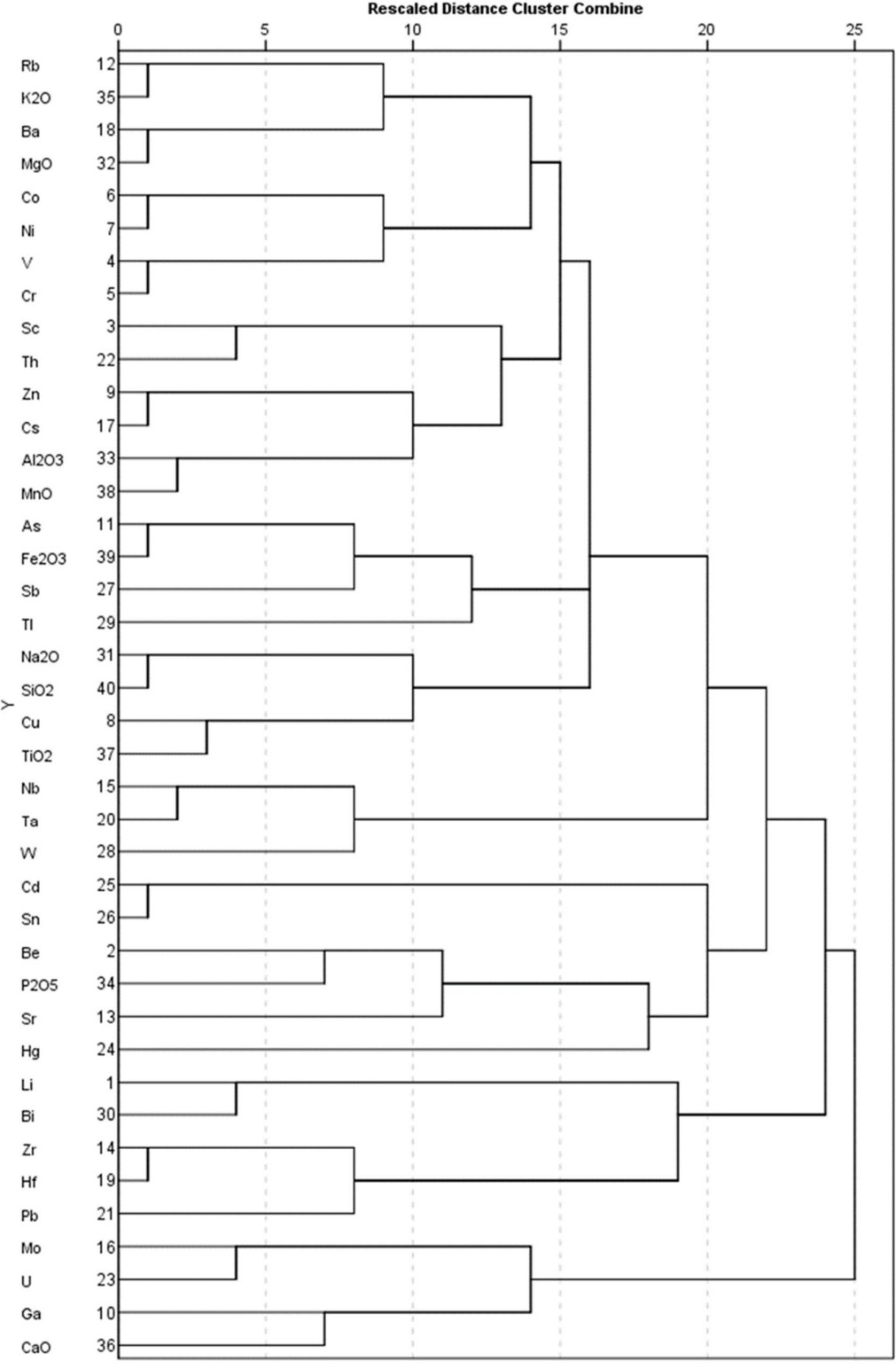

**Figure 7.** The cluster analysis of the geochemical data from the Pennsylvanian Pingshuo coal samples.

Group 1 included As, $Fe_2O_3$, Sb, $Na_2O$, $SiO_2$, Cu, and $TiO_2$. They are typical sulphophile elements, except for $Na_2O$, $SiO_2$, and $TiO_2$, which are lithophile elements.

Group 2 included Cd, Sn, Be, $P_2O_5$, Sr, and Hg. Be, $P_2O_5$, and Sr are lithophile elements while Hg is a typical sulphophile element and always coexists with As, Pb, Zn, and Sb.

Group 3 included Li, Bi, Zr, Hf, and Pb. The correlation coefficients of $SiO_2$ were all above 0.6 (Table 5), which suggested that these elements were related to silicates.

Group 4 included Mo, U, Ga, and CaO.

**Table 5.** The correlations between trace elements and major oxides.

| Trace Element | $SiO_2$ | $Al_2O_3$ | $K_2O$ | CaO | $P_2O_5$ | MgO | $Na_2O$ | $TiO_2$ | MnO | $Fe_2O_3$ | $Al_2O_3$+ $SiO_2$ |
|---|---|---|---|---|---|---|---|---|---|---|---|
| Li | 0.942 | 0.607 | 0.657 | −0.137 | −0.139 | −0.177 | 0.758 | 0.708 | −0.235 | −0.375 | −0.120 |
| Be | −0.636 | −0.652 | −0.462 | −0.133 | 0.486 | −0.165 | −0.613 | −0.542 | 0.074 | 0.196 | 0.084 |
| Sc | −0.484 | −0.021 | −0.479 | 0.705 | −0.201 | 0.582 | −0.290 | −0.614 | 0.679 | −0.113 | 0.690 |
| V | 0.024 | −0.326 | 0.275 | −0.014 | −0.228 | 0.217 | 0.160 | 0.199 | 0.174 | 0.066 | 0.773 |
| Cr | −0.429 | −0.473 | −0.251 | 0.144 | −0.139 | 0.191 | −0.365 | −0.280 | 0.338 | 0.392 | 0.682 |
| Co | −0.161 | −0.156 | −0.073 | 0.593 | −0.168 | 0.836 | 0.105 | −0.218 | 0.681 | 0.152 | 0.829 |
| Ni | −0.195 | 0.054 | −0.217 | 0.846 | −0.201 | 0.906 | 0.096 | −0.306 | 0.888 | −0.061 | 0.866 |
| Cu | 0.495 | 0.167 | 0.810 | −0.058 | 0.048 | 0.306 | 0.751 | 0.508 | −0.003 | −0.200 | 0.789 |
| Zn | −0.130 | −0.192 | −0.025 | −0.220 | −0.025 | 0.107 | −0.063 | 0.195 | −0.211 | 0.036 | 0.939 |
| Ga | −0.186 | 0.062 | −0.171 | 0.657 | −0.032 | 0.857 | 0.062 | −0.282 | 0.520 | −0.032 | 0.324 |
| As | −0.228 | −0.330 | −0.204 | −0.312 | −0.070 | −0.282 | −0.435 | −0.051 | −0.233 | 0.926 | 0.675 |
| Rb | 0.813 | 0.381 | 0.989 | −0.367 | −0.139 | −0.085 | 0.867 | 0.786 | −0.372 | −0.140 | 0.803 |
| Sr | −0.343 | −0.205 | −0.245 | −0.050 | 0.977 | −0.067 | −0.304 | −0.411 | −0.063 | −0.037 | −0.316 |
| Zr | 0.702 | 0.163 | 0.742 | −0.390 | 0.142 | 0.025 | 0.638 | 0.729 | −0.473 | −0.176 | 0.763 |
| Nb | 0.200 | 0.223 | −0.167 | 0.320 | −0.135 | 0.138 | 0.044 | −0.080 | 0.209 | −0.038 | 0.799 |
| Mo | −0.315 | −0.193 | −0.289 | 0.656 | −0.053 | 0.658 | −0.082 | −0.420 | 0.738 | 0.020 | −0.405 |
| Cs | 0.757 | 0.567 | 0.550 | −0.241 | −0.128 | −0.313 | 0.540 | 0.657 | −0.229 | 0.208 | 0.833 |
| Ba | 0.111 | 0.397 | 0.149 | 0.736 | 0.155 | 0.628 | 0.418 | −0.113 | 0.684 | −0.395 | 0.855 |
| Hf | 0.679 | 0.227 | 0.696 | −0.414 | 0.168 | −0.049 | 0.587 | 0.786 | −0.504 | −0.121 | 0.830 |
| Ta | 0.354 | 0.461 | −0.118 | 0.228 | −0.109 | −0.112 | 0.108 | 0.139 | 0.125 | −0.187 | 0.775 |
| Pb | 0.682 | 0.296 | 0.768 | −0.335 | −0.179 | 0.038 | 0.711 | 0.884 | −0.410 | −0.178 | 0.505 |
| Th | −0.488 | 0.394 | −0.503 | 0.526 | −0.078 | 0.108 | −0.345 | −0.459 | 0.470 | 0.171 | 0.659 |
| U | −0.360 | −0.179 | −0.234 | 0.648 | −0.240 | 0.795 | −0.051 | −0.384 | 0.720 | 0.016 | −0.308 |
| Ge | −0.662 | −0.591 | −0.495 | 0.092 | 0.011 | 0.233 | −0.593 | −0.541 | 0.253 | 0.753 | 0.234 |
| Cd | −0.318 | 0.275 | −0.370 | 0.278 | 0.377 | −0.044 | −0.230 | −0.298 | 0.329 | −0.037 | 0.011 |
| Sb | 0.150 | 0.303 | −0.026 | 0.572 | −0.036 | 0.772 | 0.291 | 0.023 | 0.446 | 0.136 | 0.196 |
| Tl | −0.299 | −0.399 | −0.181 | −0.288 | −0.209 | −0.282 | −0.414 | −0.059 | −0.121 | 0.850 | 0.150 |
| Bi | 0.835 | 0.674 | 0.807 | 0.029 | −0.075 | −0.036 | 0.882 | 0.649 | −0.028 | −0.494 | −0.284 |
| Hg | 0.025 | −0.412 | −0.256 | −0.547 | 0.290 | −0.586 | −0.418 | 0.021 | −0.488 | 0.447 | −0.508 |

### 5.3. Modes of Occurrence of Selected Trace Elements

5.3.1. Lithium (Li)

Lithium (Li) is a critical element that is used in batteries and other applications. Lithium enrichment has been studied by several authors [38,39]. Previous studies have shown that Li in coal is often related to aluminosilicate minerals and organic matter [38–40]. The average Li content in coal is 14 μg/g worldwide [33] and 31.8 μg/g in China [14]. The lithium contents in the Anjialing coal samples ranged from 19.3 to 181.00 μg/g, with an average of 81.3 μg/g (Table 3), which was higher than the average lithium content in other hard coal from around the world (CC = 5.81). Lithium had substantial positive correlations with $SiO_2$ and $Al_2O_3$ in the Anjialing coal (Table 5), which implied that the lithium was linked to aluminosilicate minerals in the Anjialing coal. This was also supported by the results of the sequential chemical extraction procedures.

5.3.2. Zirconium (Zr) and Hafnium (Hf)

The concentrations of Zirconium (Zr) in the Anjialing coal samples were slightly higher than the average content in Chinese coal (CC = 3.64), as were the concentrations of hafnium (Hf) (CC = 2.78). In the coal samples, Zr and Hf showed similar distributions (Figure 6) and were in the same group (Figure 7). Additionally, their correlation coefficient was 0.97, which indicated that they had similar modes of occurrence in the Anjialing coal. The strong positive correlation between $SiO_2$ and $Al_2O_3$ (Table 5) suggested that these elements were

related to aluminosilicate minerals and that clay minerals could be their primary carriers. Additionally, Zr and Hf had a strong positive correlation with $TiO_2$ and $K_2O$ (Table 5), which indicated that some Zr and Hf could occur in zircons, including anatase, and some minerals that hosted potassium.

### 5.3.3. Arsenic (As)

The concentrations of Arsenic (As) in the Anjialing coal samples were relatively low compared to the average content in other hard coal from around the world (CC = 0.1). Arsenic in the Anjialing coal was strongly correlated with $Fe_2O_3$ (0.926, Table 5) and was grouped with typical sulphophile elements (Figure 7), which indicated that As was associated with sulfide minerals, i.e., pyrite.

### 5.3.4. Lead (Pb)

The concentration of Lead (Pb) in the samples averaged 17.73 µg/g, which was two times greater than the average content in other hard coal from around the world (9 µg/g) [33]. The principal hosts for Pb in coal are galena and certain sulfide minerals, which were seen in AJL-9-07 and AJL-9-13 (Table 4). In our coal samples, the correlations between Pb and $Al_2O_3$ and $SiO_2$ were strong, as shown in (Table 5, 0.505), which suggested that Pb could also be associated with aluminosilicate minerals. This could be interpreted as the reason that Pb was grouped with Zr, Hf, and $SiO_2$ in the cluster histogram (Figure 7).

### 5.3.5. Mercury (Hg)

Mercury (Hg) is a poisonous and volatile chalcophile element [41]. Table 3 shows the concentration coefficient (CC = 3.8) of Hg in the Anjialing coal samples, which had an average Hg level of 0.38 µg/g. This average mercury content was higher than Clarke's average value for bituminous coal worldwide (0.10 µg/g) [33], even though the content of Hg in coal is usually low. High coal consumption in China has resulted in the high release of mercury from coal into the atmosphere [42]. Mercury also had positive correlation coefficients with $Al_2O_3$, $SiO_2$, and $Fe_2O_3$ (Table 5), which indicated that Hg was associated with aluminosilicate and sulfide minerals.

## 6. Conclusions

The minerals in the coal from the Anjialing coal seam 9 mainly consisted of kaolinite, boehmite, pyrite, calcite, and quartz, with occasional illite and anatase.

The lithium concentrations were considerably high in the Anjialing coal samples. The elements of Ga, Zr, Hf, U, and Hg were slightly enriched and the elements of Be, Sc, V, Cr, Cu, Zn, Sr, Nb, Mo, Ta, Pb, and Th were moderately enriched, while Co, As, Rb, Cs, Ba, Tl, and Ba were depleted.

Lithium occurred mostly in silicates. Gallium occurred more in silicate and sulfide forms in the Anjialing coal. Zirconium and Hf had strong positive correlations with clay minerals and anatase. Silicates and pyrite were the principal carriers of As in this coal seam. Mercury was associated with aluminosilicate minerals more than sulfides. Sulfide minerals, such as pyrite and carbonates, were the primary hosts and carriers of Pb.

**Author Contributions:** Conceptualization, data curation, formal analysis, methodology, software, writing—original draft, and writing—review and editing, T.J.I.; conceptualization, funding acquisition, methodology, resources, supervision, and writing—review and editing, B.S.; conceptualization, methodology, and supervision, C.L.; data curation and writing—review and editing, Y.L.; formal analysis, methodology, and software, Y.K.; methodology and writing—review and editing, X.P. All authors have read and agreed to the published version of the manuscript.

**Funding:** This work was supported by the National Science Foundation of China (grant numbers: U1810202, 41872177, 41602178, and 41802193) and the National Key R&D Program of China (grant number 2021YFC2902002).

**Data Availability Statement:** Not applicable.



**Acknowledgments:** The authors would like to thank the anonymous reviewers for their helpful suggestions and comments.

**Conflicts of Interest:** The authors declare no conflict of interest with regard to the research, authorship, and publication of this article.

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
