# Peer review of "Modes of Occurrence and Enrichment of Trace Elements in Coal from the Anjialing Mine, Pingshuo Mining District, Ningwu Coalfield, Shanxi Province, China"

_minerals, doi:10.3390/min12091082_

Round 1

Reviewer 1 Report (Previous Reviewer 2)

The manuscript was carefully revised by the authors and significant improvement was achieved. However, there is still some issues that might improve the quality of the work, and are as follow:
- Line 49-50, Literature not cited;

- Line 104, Would it be more appropriate to change "drilling core" to “borehole”?

- Line 133, The expression "water content" is not appropriate, it should be “moisture content”.

- Line 137-138, What is the basis for classifying coal? Please cite relevant literature.

- Section 4.2, The authors still do not provide data on mineral content and are asked to do so or to explain why this data is not provided.

-Line 143, “with a dominance of kaolinite and pyrite”, Why does the author say this, please provide the reasons.

- Line 181, “……. gangue (2.78) are significantly higher, ….”, Compared to what? Higher than what?

- Line 191, concentration coefficient (CC) was proposed by Dai et al. Please cite literature.

- Line 191, Would it be more appropriate to change "proportion" to “ratio”

- Line 192, world hard coal, please cite literature.

- Line 199 and 200, what is “enrichment coefficient”?

- Line 293, considerable higher than ……, Please consider this expression, the CC value is only 3.68, is it appropriate to use considerable?

- Line 318, “….., with the average Hg level of (0.38 μg/g).”

- Line 319, what is “Clarke's world bituminous coal”? Please cite literature.

Author Response

All the corrected statements in the manuscript are in "track changes" in red colors.

Comments and Suggestions for Authors

The manuscript was carefully revised by the authors and significant improvement was achieved. However, there is still some issues that might improve the quality of the work, and are as follow:

- Line 49-50, Literature not cited;

Response 1: The authors has already cited the relevant literature in the manuscript.

- Line 104, Would it be more appropriate to change "drilling core" to “borehole”?

Response 2: Based on the suggestion of the reviewer "drilling core" was changed to “borehole”.

- Line 133, The expression "water content" is not appropriate, it should be “moisture content”.

Response 3: Based on the suggestion of the reviewer "water content" was changed to “moisture content”.

- Line 137-138, What is the basis for classifying coal? Please cite relevant literature.

Response 4: The authors has already cited the relevant literature in the manuscript.

- Section 4.2, The authors still do not provide data on mineral content and are asked to do so or to explain why this data is not provided.

Response 5: We thank the reviewer’s comment, but we really did not provide the mineral content. In the study, XRD was conducted on the raw coal rather than low-temperature ash, so it is difficult to get the mineral content. In the paper, mineral composition without content were present in figure 3.

-Line 143, “with a dominance of kaolinite and pyrite”, Why does the author say this, please provide the reasons.

Response 6: Thank you for pointing out this correction to us. The authors made this conclusion according to the XRD analysis, pyrite and kaolinite were identified in almost each sample.

- Line 181, “……. gangue (2.78) are significantly higher, ….”, Compared to what? Higher than what?

Response 7: Thank you for this correction, I was supposed to compare the values of coal samples and gangue samples with the average values for Chinese coal. So, I omitted “the average values for Chinese coal (1.42)”.

- Line 191, concentration coefficient (CC) was proposed by Dai et al. Please cite literature.

Response 8: The authors have already added the relevant literature in the manuscript.

- Line 191, Would it be more appropriate to change "proportion" to “ratio”

Response 9: Based on the suggestion of the reviewer "proportion" was changed to “ratio”.

- Line 192, world hard coal, please cite literature.

Response 10: The authors have already added the relevant literature in the manuscript.

- Line 199 and 200, what is “enrichment coefficient”?

Response 11: Thanks for pointing it out to me, I was meant to write “concentration coefficient” instead of “enrichment coefficient”.

- Line 293, considerable higher than ……, Please consider this expression, the CC value is only 3.68, is it appropriate to use considerable?

Response 12: I acknowledged your correction, and I consider changing “considerably” to “slightly”.

- Line 318, “….., with the average Hg level of (0.38 μg/g).”

Response 13: Thanks for the suggestion, “with the average Hg level of (0.38 μg/g)” was added to the sentence.

- Line 319, what is “Clarke's world bituminous coal”? Please cite literature.

Response 14: The authors have already cited the relevant literature in the manuscript.

Reviewer 2 Report (Previous Reviewer 3)

The manuscript has been largely revised following the comments and suggestion of reviewers. I agree the acception of this manuscript.

Author Response

Dear Highly Esteemed Reviewer,

We thanks you for the acceptance of this manuscript.

Regards

Ikeh

This manuscript is a resubmission of an earlier submission. The following is a list of the peer review reports and author responses from that submission.

Round 1

Reviewer 1 Report

Dear Editors and Authors

I reviewed the significant improved version of the MS and I am happy to say that the authors indeed provide with a better MS. 

There is still few issues that must be adressed.

1. In results please do not add data that you did not collect. If from literature it must be pointed before or in the discussion. 

 2. Please add the basis (i.e. as received, dry basis etc.)  on various parameters.

3. Introduction is still local and not appropriate for International Journal.

4. Some more comments in the attached pdf.

Reviewer 2 Report

This revised version has some qualitative improvements over the previous version, but some basic and important issues have not been raised, particularly those raised by the reviewer, and have not been answered. The author seems to have paid little attention to the reviewer's comments, and only gave the answer that the reviewer's comments had been revised, but did not give a detailed answer on how to revise them. This article must be majorly revised before it can be reconsidered for publication in this journal.

Reviewer 3 Report

 Comments and Suggestions for Authors

Manuscript ID: minerals-1753582

Title: Mode of occurrence and Enrichment of Trace Elements in Coal 2 from Anjialing Mine, Pingshuo coal mine, Ningwu Coalfield, 3 Shanxi Province, China

Authors: Tobechukwu Justin Ikeh  et al.

The manuscript has revised in some parts, and improved some in quanlity. However, there are some pitfalls. Firstly, the English writing should be improved by English editing company. Secondly, there are some problems attached in the following senctences. Some problems indicated previously are not solved in the revised manuscript. The article can be accepted in this journal after major revision.

The specific suggestions are the following:

(1) In section 3.0 Materials and Methods-XRD analysis, the authors recorded XRD over a 2θ interval of 10-80°. In fact, there are some diffraction peaks of clay minerals (such as illite, chlorite, smectite) on diffraction region between 5-10°. In your manuscript, illite is identified but the diffraction peaks (2θ=8.7°) are absent. How do you known the existence of illite? The diffraction peaks (2θ=8.7°) are missing, how do you calculate the contents?

(2) In table 2 and Figure 3, the authors identify the occurrence of muscovite. In fact, the muscovite is illite because muscovite and illite have the same diffraction peaks. Additionally, muscovite is a very easily decomposed mineral in acidic peat swamp and can be altered to clay minerals such as illite. The decisive problem is that the peaks<10°are missing, the illite cannot identified.

(3) The absence of mineral conent can be supplemented. Please provide mineral content.

(4) In table 3, the major elements are conducted by ICP-OES. The silicon is not determined by ICP-OES. Thus, how do you get the content of SiO2 in table 3.

(5) In figure 3, the sample AJL-9-R is listed, but it is missing in table 2, 3. Is the sample AJL-9-R is the roof? The roof contains highest Fe abundance but pyrite is not found in this samples. How do you explain this result? Sample AJL- 9-03G, 9-13, and floor show the existence of pyrite in Figure 3 and contain relatively high Fe content. However, the sample roof with highest iron abundance displays no pyrite. Where is iron from?